# Factors associated with the job satisfaction of certified nurses and nurse specialists in cancer care in Japan: Analysis based on the Basic Plan to Promote Cancer Control Programs

**Masaki Kitajima[1], Chiharu Miyata[2], Keiko Tamura[1], Ayae Kinoshita[1], Hidenori Arai[3] \***

**1** Human Health Sciences, Graduate School of Medicine, Kyoto University, Kyoto, Japan, **2** Nursing, Mie University Graduate School of Medicine, Mie, Japan, **3** National Center for Geriatrics and Gerontology, Obu, Aichi, Japan

\* harai@ncgg.go.jp

**Data Availability Statement:** The minimal anonymized dataset necessary to replicate our

## Abstract

### Background

As the Japanese population ages, the number of cancer patients will likely increase. Therefore, qualified cancer health care providers should be recruited and retained. Nurse job satisfaction is influenced by numerous factors and may affect staff turnover and patient outcomes.

### Objectives

To evaluate the job satisfaction of certified nurses and nurse specialists in Japanese cancer care and elucidate factors associated with job satisfaction.

### Methods

Participants in this cross-sectional study comprised 200 certified nurse specialists and 1,472 certified nurses working in Japanese cancer care. A chi-square test and logistic regression analysis were conducted to identify job satisfaction factors.

### Results

Job satisfaction was present in 38.45% and 49.00% of certified nurses and nurse specialists, respectively. Certified nurses associated job satisfaction with cross-departmental activities (OR 2.24, $p<0.001$), positive evaluation from senior stuff (OR 4.58, $p<0.001$), appropriate staff allocation (OR 1.75, $p<0.001$), more than five years certified nurse experience (OR 1.91, $p<0.001$), and positive evaluation of the development of certified nurses (OR 2.13, $p<0.01$) and nurse specialists (OR 1.37, $p<0.05$). Low job satisfaction was associated with working on a ward (OR 0.51, $p<0.001$) and a capacity of more than 200 beds (OR 0.33, $p = 0.00$). Certified nurse specialists associated job satisfaction with palliative care

study is available within the Supporting Information files.

**Funding:** This study was supported by research grants from The Japan Geriatrics Society (H.A.), JSPS KAKENHI (JP17K18202) (M.K.) and Yasuda Memorial Medical Foundation (M.K.). The funders had no role in study design, data collection and analysis, decision to publish, or preparation of the manuscript.

**Competing interests:** No authors have competing interests.

team participation (OR 2.64, $p<0.05$), cross–sectional activities (OR 7.06, $p<0.01$), positive evaluation from senior stuff (OR 13.15, $p<0.001$), presence of certified nurses in radiation therapy (OR 2.91, $p<0.05$), positive certified nurse specialist development evaluation (OR 7.35, $p<0.001$), medical service fees (OR 3.78, $p<0.01$), and independent activities (OR 11.34, $p<0.01$).

## Conclusions

We identified factors related to activities, facilities, and the cancer care team associated with job satisfaction of certified nurses and nurse specialists in Japanese cancer care. Suggestions are provided to enhance job satisfaction through Japan's Basic Plan to Promote Cancer Control, which may help hospital administrators retain nursing staff.

## Introduction

With the aging of the global population, the number of cancer patients and cancer-related deaths in Japan is expected to increase [1]. To provide high-quality medical services and optimally support cancer patients and their families, the Basic Plan to Promote Cancer Control Programs was adopted by the Japanese national health care system. It suggests a collaborative approach involving multiple healthcare professionals.

It is important to ensure a sufficient staff of specialized care providers including nursing professionals with qualifications awarded by the Japan Nursing Association [1]. A certified nurse specialist (CNS) is a nurse who participates in clinical practice, consultation, coordination of activities, ethical management, education, and research; has at least five years of nursing experience; and has obtained certification after completing a master's degree at a graduate school. Through their job activities, CNSs aim to improve the quality of medical care, propose policies, and work to maintain and improve patient health [2]. Therefore, they are often assigned to nursing administration offices and frequently involved in activities to improve the entire facility, such as staff education, research, and patient discharge.

A certified nurse (CN) is a nurse who participates in clinical practice, teaching, and consultation; has more than five years of nursing experience; and has obtained certification after graduating from a special vocational school approved by the Japan Nursing Association. CNs are generally assigned as staff nurses in a single department such as an outpatient clinic or a ward [3].

In accordance with the stipulations of the Basic Plan to Promote Cancer Control, CNSs and CNs are recruited to hospitals authorized to treat cancer to alleviate the burden on cancer patients and their families and improve the quality of recuperation. Furthermore, CNSs and CNs are assigned to the palliative care team and provide patient counseling. Because of the critical role of oncology CNSs and CNs in patient care, understanding the factors influencing their job satisfaction and performance has important implications for clinical practice. Since the Basic Plan to Promote Cancer Control stipulates the most important measures in the treatment and support of cancer patients, it is important to incorporate it in the evaluation of the job satisfaction of CNs and CNSs [4].

The job satisfaction of nurses has been addressed in other settings [5–10]. A recent integrative review showed that job satisfaction among nurses is a variable and complex phenomenon depending on numerous factors [7]. The factors varied between studies and included adequate staffing and equipment, job security and compensation, opportunities for professional

development, supervisor support, autonomy, quality of workplace relationships, and the feeling that one's job makes a difference.

Many studies evaluated the professional growth, role, support, and recognition of CNSs and CNs [11–21]. However, research on job satisfaction has only been conducted in the fields of psychiatric mental health nursing for CNSs and in dysphagia nursing for CNs [22–25]. CNSs are nursing specialists whose role requires substantial autonomy. Reportedly, the autonomous role of a caregiver improves the job satisfaction and performance of nurses [26–30]. Furthermore, the concept of a specialist implies autonomy including responsibility and expertise [31].

No study has evaluated job satisfaction among CNSs and CNs in the field of oncology in Japan. However, this information is crucial in the retention of nursing stuff and provision of optimal patient care, since job satisfaction presumably affects the expertise of CNSs and CNs and their ability to provide outstanding care. In addition, the specific job profiles and responsibilities of CNSs and CNSs may be associated with distinct job satisfaction factors. Identifying these factors may facilitate the implementation of specific measures to improve job satisfaction. Therefore, we assessed the job satisfaction of CNSs and CNs working in cancer care in Japan and identified factors enabling them to operate productively in their organizations.

## Methods

### Survey development and testing

Because CNSs and CNs have unique roles, existing scales could not be applied in this study. Previous reports were used to identify possible factors influencing job satisfaction for inclusion in the current investigation [22–25]. Furthermore, the Basic Plan to Promote Cancer Control was utilized to select questions pertaining to personnel deployment requirements for CNSs and CNs, facility types, facility requirements, medical care remuneration, and palliative care team placement requirements.

Items were divided into categories related to activities, facilities, and participation in the cancer care team. To examine the content validity of the questions, university faculty members well acquainted with the Basic Plan to Promote Cancer Control and activities of CNSs and CNs discussed and agreed on the included items. Next, a pretest was administered to four CNSs and four CNs, who were asked whether they faced any difficulties in answering the questions and understanding the terms used. The terminology was revised as necessary. Four researchers extracted the characteristic activities of the CNSs and CNs based on the Basic Plan to Promote Cancer Control. The extracted items and question items on job satisfaction were examined, and the relevance of each was investigated.

### Job satisfaction questionnaire: Design and content

The following are examples of questions asked: "Are cross-sectional activities conducted in the hospital?" "Is the personnel placement of the assigned department appropriate?" "Do you feel that you are well evaluated by your boss?" Furthermore, eleven questions concerned factors related to activities (including demographics), seven concerned factors related to facilities, and four concerned factors related to the cancer care team. Table 1 presents the questions. The study participants were asked to respond to the questionnaire survey on a simple three-point scale, which was used because of the difficulty in securing time during working hours in a busy clinical practice. The pretest was measured on a five-point scale, but it required extended time to answer and appeared to cause a significant clinical burden. A confirmatory factor analysis (CFA) was conducted to verify the factor structure of the questionnaire. A three-factor model was analyzed, including factors related to activities, facilities, and the cancer team. The following values were obtained for the model fit: Comparative fit index (CFI) 0.96, Tucker–Lewis

**Table 1. Contents of the questionnaire.**

| |
|---|
| 1. Factors related to activities (including demographics) |
| ① Years of experience after acquiring nursing license (open-ended) <br> ② Years of experience after acquiring CNS or CN qualifications (open-ended) <br> ③ The presence or absence of other CNSs and CNs working in the same institution (open-ended) |
| ④ Working system (multiple answers)[1] (palliative care team, cancer consultation and support center, outpatient chemotherapy room, radiotherapy room, outpatient department, ward department) |
| ⑤ Position (open-ended) |
| ⑥ Lecturer for study group or training (frequently and regularly; yes, but not that frequently; no) |
| ⑦ Experienced the launch of a department or patient group (yes, no) |
| ⑧ Opportunity for cross-departmental activities[1] (always; not always, but regularly; never) |
| ⑨ Opportunity for exchange with CNSs and CNs in other institutions (regularly; yes, but not regularly; never) |
| ⑩ Positive evaluation from senior staff (yes, sometimes, never) <br> ⑪Appropriate staff allocation |
| 2. Factors related to facilities |
| ① Type of hospital (prefectural cancer center hospital, regional cooperation cancer center hospital, cancer center hospital designated by prefecture, community hospital, other) |
| ② Bed capacity (0 to 199, 200 to 499, $\geq$ 500, other) |
| ③ Publicize information about the existence of CNSs and CNs among community members and the public (yes, no) |
| ④ Service system in the following four departments pertaining to cancer(chemotherapy room, palliative care team, cancer consultation and support center, radiotherapy room) |
| ⑤ Additional medical service compensation for cancer <br> (palliative care practice addition, cancer patient counseling charges, outpatient palliative care management charges, cancer pain palliation instruction charges) |
| ⑥ Implementation of cancer care in the institution <br> (based on the requirements of designated cancer centers and hospitals) |
| ⑦ Positive evaluation of CNS and CN development in the institution (always, sometimes, never) |
| 3. Factors related to the cancer care team |
| ① Coordination among multiple healthcare professionals (always, sometimes, never) |
| ② Independent activities[2] (always, sometimes, never) |
| ③ Availability of conferences (times per week) <br> ④ Job type (pharmacist, medical social worker, nutritionist, physical therapist, occupational therapist, speech therapist, clinical psychologist) |

Regarding department affiliations, if individuals were involved with two or more affiliations concurrently, they were instructed to choose multiple answers.

Refers to the ability to fulfill one's role without interference from others.

index (TLI) 0.90, and root mean-square error of approximation (RMSEA) 0.04. Overall, the findings of the CFA indicated an acceptable fit of the three-factor model. Furthermore, the reliabilities of the three scales were estimated by calculating Cronbach's alpha, which was 0.79, 0.81, and 0.72 for factors related to activities, facilities, and the cancer team, respectively.

In addition, we enquired about job satisfaction. We asked the question, "Are you satisfied with your job?" Originally, we used a three-level scale with the following response options: "satisfied," "somewhat satisfied," and "dissatisfied." However, upon review, we combined the first two levels to yield a two-level scale: "satisfied" and "dissatisfied."

## Procedure

This study was cross-sectional and involved the use of self-report questionnaires. A survey request, self-report questionnaire, document explaining the implications of the study, and

return envelope were mailed to each participant. The recipients of the survey were asked to return the questionnaire to the researchers within two months. The survey period was between May and July 2014. In addition, a written notification was included informing the participants that their participation was voluntary. Precautions were taken to ensure that the department heads were unaware of whether the CNSs and CNs had mailed the completed questionnaires.

## Participants

In February 2014, 3,450 nurses were included on the list of registered CNSs and CNs on the official website of the Japan Nursing Association. Of these, 3,332 nurses (483 CNSs and 2,849 CNs) whose addresses could be confirmed were mailed the questionnaire. In total, 1,696 nurses working in cancer centers or hospitals involved in cancer care responded to the mailed questionnaire. The valid response rate was 98.60% (1,672 respondents).

## Ethical considerations

The study protocol was approved by the Institutional Ethics Committee of the Kyoto University Graduate School of Medicine and Faculty of Medicine Hospital with protocol number E2072. The purpose of the study, benefits of participation, confirmation of voluntary participation, and an assurance that all data obtained in the present study would be used only for scientific purposes and anonymously were communicated to all participants in written form. All participants gave their informed consent. To protect confidentiality, personal data were kept separately from the completed questionnaires, which were coded. The study adhered to the STROBE checklist.

## Data analysis

Descriptive statistics were computed for each questionnaire item. The three-point scale had the anchors *Carried out*, *Somewhat carried out*, and *Not carried out*. Responses were categorized in a binary format, in which *Carried out* and *Somewhat carried out* represented an affirmative *Yes* response (1), and *Not carried out* represented a *No* response (2). In addition, a "bed capacity" of 0–199 was assigned a score of 1, and a capacity $\geq$ 200 a score of 2. For the type of hospital, a cancer center was assigned a score of 1, and "other hospital" a score of 2. Conferences were scored 1 for "Held/Yes" or 2 for "Not held/No."

A chi-square test was performed to evaluate the relationship between job satisfaction and other variables. Factors statistically significant in the univariate analysis were included in the subsequent multivariable analysis. To identify factors related to job satisfaction, a logistic regression analysis was conducted with satisfaction or dissatisfaction as the dependent variable, other factors as the explanatory variables, and years of experience after acquiring a nursing license as the control variable. The following explanatory variables were used in the case of categorical variables. A dummy variable was created that was assigned a value of 0 in the case of "Other hospital" for "type of hospital," "0–199" for "bed capacity," and "0–4 years" for "years of experience after obtaining CN or CNS qualification." SPSS Statistics 21 software (IBM-SPSS, Inc., Chicago, IL, USA) was used for all analyses. Two-tailed tests were performed with an alpha level of .05. Because of the exploratory nature of the study, no correction for multiple testing was applied.

## Results

### Participant characteristics

The participants were 200 CNSs and 1,472 CNs with a mean length of clinical experience of 19.8 years. Job satisfaction was present in 38.45% of CNs and 49.00% of CNSs. In terms of workplace, 71.50% of the participants worked at designated cancer centers or hospitals, and the remaining ones

**Table 2. Demographic characteristics.**

| | | CNS | | CN | |
|---|---|---|---|---|---|
| | | *N* | % | *N* | % |
| Job satisfaction | Presence Absence | 98 102 | 49.00 51.00 | 566 906 | 38.45 61.55 |
| Clinical nursing experience (years) | 5–9 | 25 | 12.50 | 93 | 6.30 |
| | 10–15 | 48 | 24.00 | 354 | 24.00 |
| | 16–20 | 71 | 35.50 | 402 | 27.30 |
| | 21–25 | 32 | 16.00 | 353 | 24.00 |
| | 26–30 | 20 | 10.00 | 194 | 13.20 |
| | 31–35 | 4 | 2.00 | 66 | 4.50 |
| | 36–35 | 0 | 0.00 | 7 | 0.50 |
| | 36–40 | 0 | 0.00 | 2 | 0.10 |
| | 41–45 | 0 | 0.00 | 1 | 0.10 |
| Clinical experience as a CNS/CN (years) | Less than 5 | 148 | 74.00 | 895 | 60.80 |
| | 5 or more | 52 | 26.00 | 577 | 39.20 |
| Affiliation | Yes | 82 | 41.00 | 409 | 27.80 |
| Palliative care team | No | 118 | 59.00 | 1064 | 72.20 |
| Cancer consultation and support center | Yes | 54 | 27.00 | 83 | 5.60 |
| | No | 146 | 73.00 | 1390 | 94.40 |
| Chemotherapy room | Yes | 26 | 27.00 | 402 | 27.30 |
| | No | 174 | 73.00 | 1071 | 72.70 |
| Radiotherapy room | Yes | 4 | 2.00 | 65 | 4.40 |
| | No | 196 | 98.00 | 1408 | 95.60 |
| Outpatient department | Yes | 28 | 14.00 | 356 | 24.20 |
| | No | 172 | 86.00 | 1117 | 75.80 |
| Ward department | Yes | 59 | 29.50 | 629 | 42.70 |
| | No | 141 | 70.50 | 1404 | 57.30 |
| Position | Manager | 121 | 60.50 | 629 | 42.70 |
| | Staff | 79 | 39.50 | 843 | 57.30 |
| Workplace | Cancer centers | 175 | 87.50 | 1021 | 69.40 |
| | Other hospitals | 25 | 12.50 | 451 | 30.60 |
| Bed capacity | Less than 200 | 9 | 4.50 | 165 | 11.20 |
| | 200 or more | 191 | 95.50 | 1307 | 88.80 |

CNS: *n* = 200, CN: *n* = 1472

worked at other hospitals (e.g., general hospitals or clinics) also involved in cancer care. Table 2 details the number of nurses working in different types of facilities or having a position title.

## Univariate analysis

The results of the chi-square test showed significant differences in the scores for 21 items for CNSs and 32 items for CNs (Tables 3 and 4). For CNSs, 11 differences were associated with activities, 7 with facilities, and 3 with the cancer care team. For CNs, 15 differences were associated with activities, 12 with facilities, and 5 with the cancer care team.

## Multivariable analysis

The results of the chi-square test were used in a logistic regression analysis to identify variables particularly important to the job satisfaction of CNSs and CNs. Table 5 presents the results of this analysis.

**Table 3. Univariate analyses for CNSs.**

| | | N | Satisfaction n (%) | Dissatisfaction n (%) | P |
|---|---|---|---|---|---|
| **Factors related to activity** | | | | | |
| Clinical nursing experience (years) | Less than 19 | 105 | 48 (45.70) | 57 (54.30) | 0.33 |
| | 19 or more | 95 | 50 (52.60) | 45 (47.40) | |
| Years of experience after acquiring CNS qualification (see Note) | Less than 5 | 148 | 62 (47.90) | 86 (58.10) | 0.00 |
| | 5 or more | 52 | 36 (69.20) | 16 (30.80) | |
| Presence in the same facility | Yes | 115 | 46 (40.00) | 69 (60.00) | 0.16 |
| • CNS | No | 85 | 36 (42.40) | 79 (57.60) | |
| • CN in palliative care | Yes | 150 | 75 (50.00) | 75 (50.00) | 0.62 |
| | No | 50 | 23 (46.00) | 27 (54.00) | |
| • CN in cancer pain management | Yes | 114 | 55 (48.20) | 59 (51.80) | 0.81 |
| nursing | No | 86 | 43 (50.00) | 43 (50.00) | |
| • CN in cancer | Yes | 163 | 81 (49.70) | 82 (50.30) | 0.72 |
| chemotherapy nursing | No | 37 | 17 (45.90) | 20 (54.10) | |
| • CN in radiation therapy | Yes | 66 | 35 (53.00) | 31 (47.00) | 0.04 |
| nursing | No | 134 | 63 (47.00) | 71 (53.00) | |
| • CN in breast | Yes | 74 | 35 (47.30) | 39 (52.70) | 0.71 |
| cancer nursing | No | 126 | 63 (50.00) | 63 (50.00) | |
| Affiliation | Yes | 79 | 47 (59.50) | 32 (40.50) | 0.02 |
| • Palliative care team | No | 121 | 51 (42.10) | 70 (57.90) | |
| • Cancer consultation and support center | Yes | 54 | 35 (64.80) | 19 (35.20) | 0.01 |
| | No | 146 | 36 (24.70) | 83 (75.30) | |
| • Chemotherapy clinic | Yes | 26 | 12 (46.20) | 14 (53.80) | 0.76 |
| | No | 174 | 86 (49.40) | 88 (50.60) | |
| • Radiotherapy room | Yes | 4 | 3 (75.00) | 1 (25.00) | 0.29 |
| | No | 196 | 95 (48.50) | 101 (51.50) | |
| • Outpatient department | Yes | 28 | 13 (46.40) | 15 (53.60) | 0.77 |
| | No | 172 | 85 (49.40) | 87 (50.60) | |
| • Ward department | Yes | 59 | 13 (22.00) | 46 (78.00) | 0.00 |
| | No | 141 | 85 (60.30) | 56 (39.70) | |
| Position | Manager | 121 | 73 (60.30) | 48 (39.70) | 0.00 |
| | Staff | 79 | 25 (31.60) | 54 (68.40) | |
| Role in educating others | Yes | 185 | 96 (51.90) | 89 (48.10) | 0.00 |
| | No | 15 | 2 (13.30) | 13 (86.70) | |
| Launched a department or patient group | Yes | 79 | 26 (32.90) | 53 (67.10) | 0.00 |
| | No | 121 | 72 (59.50) | 49 (40.50) | |
| Cross-departmental activities | Yes | 151 | 93 (61.60) | 58 (38.40) | 0.00 |
| | No | 49 | 5 (10.20) | 44 (89.80) | |
| Information exchanges or joint workshops etc. with other facilities | Yes | 169 | 89 (52.70) | 80 (47.30) | 0.02 |
| | No | 31 | 9 (29.00) | 22 (71.00) | |
| High ratings from senior staff | Yes | 150 | 95 (63.30) | 55 (36.70) | 0.00 |
| | No | 50 | 3 (6.00) | 47 (94.00) | |
| Appropriate staff allocation | Yes | 86 | 53 (61.63) | 33 (38.37) | 0.02 |
| | No | 114 | 45 (39.47) | 69 (60.53) | |
| **Factors related to facilities** | | | | | |
| Type of hospital | Cancer centers | 175 | 88 (50.30) | 87 (49.70) | 0.34 |
| | Other hospitals | 25 | 10 (40.00) | 15 (60.00) | |

(*Continued*)

**Table 3.** (*Continued*)

| | | N | Satisfaction n (%) | Dissatisfaction n (%) | P |
|---|---|---|---|---|---|
| Bed capacity | Less than 200 | 9 | 3 (33.30) | 6 (66.70) | 0.40 |
| | 200 or more | 191 | 95 (49.70) | 96 (50.30) | |
| Advertisements for CNSs and CNs | Yes | 185 | 91 (49.20) | 71 (50.80) | 0.85 |
| | No | 15 | 7 (46.70) | 8 (53.30) | |
| Service system in four departments | Yes | 152 | 81 (53.30) | 71 (46.70) | 0.03 |
| | No | 48 | 17 (35.40) | 31 (64.60) | |
| Additional medical services Palliative care practice | Yes | 111 | 58 (52.30) | 53 (47.70) | 0.30 |
| | No | 89 | 40 (44.90) | 49 (55.10) | |
| • Cancer patient counseling | Yes | 139 | 75 (54.00) | 64 (46.00) | 0.45 |
| | No | 61 | 23 (37.70) | 38 (62.30) | |
| • Outpatient palliative care management | Yes | 74 | 42 (56.80) | 32 (43.20) | 0.09 |
| | No | 126 | 56 (44.40) | 70 (55.60) | |
| • Cancer pain palliation instructions | Yes | 140 | 80 (57.10) | 60 (42.90) | 0.00 |
| | No | 60 | 18 (30.00) | 42 (70.00) | |
| Requirements • Second opinion | Received | 186 | 94 (53.40) | 82 (46.60) | 0.11 |
| | Not received | 14 | 4 (28.60) | 10 (71.40) | |
| • Collaboration team and the primary care physician | Received | 99 | 59 (59.60) | 40 (40.40) | 0.00 |
| | Not received | 101 | 39 (38.60) | 62 (61.40) | |
| • Operation of cancer regional alliances path | Received | 187 | 96 (51.30) | 91 (48.70) | 0.00 |
| | Not received | 13 | 2 (15.40) | 11 (84.60) | |
| • Training of local healthcare professionals | Received | 171 | 86 (50.30) | 85 (49.70) | 0.01 |
| | Not received | 29 | 12 (41.40) | 17 (58.60) | |
| • Hospital cancer registry | Received | 167 | 86 (51.50) | 81 (48.50) | 0.11 |
| | Not received | 33 | 12 (36.40) | 21 (63.60) | |
| Education or support in obtaining CNS qualification of the assigned facilities | Yes | 99 | 70 (70.70) | 29 (29.30) | 0.00 |
| | No | 101 | 28 (27.70) | 73 (72.30) | |
| Education or support in obtaining CN qualification of the assigned facilities | Yes | 156 | 92 (59.00) | 64 (41.00) | 0.00 |
| | No | 44 | 6 (13.60) | 38 (86.40) | |
| **Factors related to the cancer care team** | | | | | |
| Coordination among healthcare professionals | Yes | 194 | 98 (50.50) | 96 (49.50) | 0.00 |
| | No | 6 | 0 (0.00) | 6 (100.00) | |
| Independent activities | Yes | 98 | 96 (98.00) | 2 (2.00) | 0.00 |
| | No | 102 | 62 (60.80) | 40 (39.20) | |
| Availability of conferences | Yes | 187 | 96 (51.30) | 91 (48.70) | 0.01 |
| | No | 13 | 2 (15.40) | 11 (84.60) | |
| Job type• Pharmacist | Involvement | 183 | 90 (49.20) | 93 (50.80) | 0.44 |
| | Non-involvement | 17 | 8 (47.10) | 9 (52.90) | |
| • Medical social worker | Involvement | 175 | 86 (49.10) | 89 (50.90) | 0.92 |
| | Non-involvement | 25 | 12 (48.00) | 13 (52.00) | |
| • Nutritionist | Involvement | 140 | 66 (47.10) | 74 (52.90) | 0.42 |
| | Non-involvement | 60 | 32 (53.30) | 28 (46.70) | |
| • Physical therapist | Involvement | 132 | 61 (46.20) | 71 (53.80) | 0.27 |
| | Non-involvement | 68 | 37 (54.40) | 31 (45.60) | |
| • Occupational therapist | Involvement | 85 | 38 (44.70) | 47 (55.30) | 0.30 |
| | Non-involvement | 115 | 60 (52.20) | 55 (47.80) | |
| • Speech therapist | Involvement | 54 | 21 (38.90) | 33 (61.10) | 0.08 |
| | Non-involvement | 146 | 77 (52.70) | 69 (47.30) | |

(*Continued*)

**Table 3.** (Continued)

|  |  | N | Satisfaction n (%) | Dissatisfaction n (%) | P |
|---|---|---|---|---|---|
| • Clinical psychologist | Involvement | 103 | 55 (53.40) | 48 (46.60) | 0.20 |
|  | Non-involvement | 97 | 43 (44.30) | 54 (55.70) |  |

The threshold for years of experience was set at 5 years, because according to [32], 44.00% of CNs are hesitant to renew their 5-year contracts the first time. Reasons given for their hesitation include: "There is no satisfaction in the work" and they have "not been able to fully refresh."

**Multivariable analysis of job satisfaction in CNSs.** In terms of the factors related to activities, job satisfaction was present when the participant belonged to the palliative care team [odds ratio (OR) = 2.64], cross-departmental activities could be performed (OR = 7.06), the participant received a favorable rating from senior staff (OR = 13.15), and a CN was involved in radiation therapy nursing in the same institution (OR = 2.91). Of the factors related to facilities, job satisfaction was found when there was a high rating for CNS development in the institution (OR = 7.35) and the participant belonged to an institution where an additional pain relief management fee was charged (OR = 3.78). Of the factors related to the cancer care team, job satisfaction was found when independent activities could be performed (OR = 11.3) (Cox-Snell $R^2$ = 0.49; Nagelkerke $R^2$ = 0.65).

**Multivariable analysis of job satisfaction in CNs.** Of the factors related to activities, working on a ward was associated with the absence of job satisfaction (OR = 0.49). In contrast, job satisfaction was present when cross-departmental activities could be performed (OR = 2.24), a high rating was received from senior staff (OR = 4.88), there was appropriate staff allocation in each ward or department (OR = 1.75), and the participant had at least five years of experience after acquiring a CN qualification (OR = 1.91). Of the factors related to facilities, job satisfaction was present when the capacity was less than 200 beds (OR = 0.33) and there was a high rating of CNS and CN development in the institution (OR = 1.37 and 2.13, respectively). Of the factors related to the cancer team, job satisfaction was present when independent activities could be performed (OR = 6.83) (Cox-Snell $R^2$ = 0.33; Nagelkerke $R^2$ = 0.45).

## Discussion

In this study on CNs and CNSs working in cancer care in Japan, we identified numerous factors related to activities, facilities, or the cancer team that influenced job satisfaction.

Of the factors related to activities, opportunities for cross-departmental activities and positive evaluation from senior stuff were common to CNs and CNSs.

In our study, CNSs were often affiliated with departments engaged in cross-departmental activities throughout the entire facility (Fig 1).

Cross-departmental activities increased the job satisfaction of CNSs and CNs. However, these mainly involved activities by the palliative care team and the counseling and support center. In this study, only 409 members of palliative care teams and 83 members of the counseling and support centers were surveyed, and a high number of respondents (1,404) were affiliated with a single department (Fig 2). In facilities without CNSs, consultations in other wards and care for patients during radiotherapy require specialized abilities and offer CNs an opportunity to demonstrate their professional skills, which is thought to increase job satisfaction [33, 34].

Our findings are consistent with those of a previous study that showed that consciously conducting professional activities as a specialist increases work engagement, which influences job satisfaction [35]. Furthermore, the feeling that a job is worthwhile and that one's

**Table 4. Univariate analyses for CNs.**

| | | N | Satisfied n (%) | Not satisfied n (%) | P |
|---|---|---|---|---|---|
| **Factors related to activity** | | | | | |
| Clinical nursing experience (years) | Less than 19 | 849 | 320 (37.70) | 529 (62.30) | 0.48 |
| | 19 or more | 623 | 246 (39.50) | 377 (60.50) | |
| Years of experience after acquiring CN qualification (see Note) | Less than 5 | 895 | 270 (30.20) | 625 (69.80) | 0.00 |
| | 5 or more | 577 | 296 (51.30) | 281 (48.70) | |
| Presence in the same facility• CNS | Yes | 467 | 199 (42.60) | 268 (57.40) | 0.02 |
| | No | 1005 | 367 (36.50) | 638 (63.50) | |
| • CN in palliative care | Yes | 962 | 390 (40.50) | 572 (59.50) | 0.02 |
| | No | 510 | 176 (34.50) | 334 (65.50) | |
| • CN in cancer pain management nursing | Yes | 603 | 233 (38.60) | 370 (61.40) | 0.74 |
| | No | 869 | 333 (38.30) | 536 (61.70) | |
| • CN in cancer chemotherapy nursing | Yes | 911 | 374 (41.10) | 537 (58.90) | 0.04 |
| | No | 561 | 192 (34.20) | 369 (65.80) | |
| • CN in radiation therapy nursing | Yes | 310 | 137 (44.20) | 173 (55.80) | 0.01 |
| | No | 1162 | 429 (37.00) | 733 (63.00) | |
| • CN in breast cancer nursing | Yes | 363 | 158 (43.50) | 205 (56.50) | 0.03 |
| | No | 1109 | 408 (36.80) | 701 (63.20) | |
| Affiliation | Yes | 409 | 186 (45.50) | 223 (54.50) | 0.00 |
| • Palliative care team | No | 1063 | 380 (35.70) | 683 (64.30) | |
| • Cancer consultation and support center | Yes | 83 | 45 (54.20) | 38 (45.80) | 0.00 |
| | No | 1389 | 521 (37.50) | 868 (62.50) | |
| • Chemotherapy room | Yes | 402 | 167 (41.50) | 235 (58.50) | 0.14 |
| | No | 1070 | 399 (37.30) | 671 (62.70) | |
| • Radiotherapy room | Yes | 65 | 19 (29.30) | 46 (70.70) | 0.12 |
| | No | 1407 | 547 (38.90) | 860 (61.10) | |
| • Outpatient department | Yes | 356 | 130 (36.50) | 226 (63.50) | 0.39 |
| | No | 1116 | 436 (39.10) | 680 (60.90) | |
| • Ward department | Yes | 629 | 177 (28.10) | 452 (71.90) | 0.00 |
| | No | 843 | 389 (46.10) | 454 (53.90) | |
| Position | Manager | 629 | 177 (28.10) | 452 (71.90) | 0.00 |
| | Staff | 843 | 389 (46.10) | 454 (53.90) | |
| Role in teaching other nurses | Yes | 1359 | 533 (39.20) | 826 (60.80) | 0.00 |
| | No | 113 | 33 (29.20) | 80 (70.80) | |
| Launched a department or patient group | Yes | 877 | 305 (34.80) | 572 (65.20) | 0.00 |
| | No | 595 | 261 (43.90) | 334 (56.10) | |
| Cross-departmental activities | Yes | 1053 | 500 (47.50) | 553 (52.50) | 0.00 |
| | No | 419 | 66 (15.80) | 353 (84.20) | |
| Information exchanges or joint workshops etc. with other facilities | Yes | 1209 | 505 (41.80) | 704 (58.20) | 0.00 |
| | No | 263 | 61 (23.20) | 202 (76.80) | |
| High ratings from senior staff | Yes | 980 | 511 (52.10) | 469 (47.90) | 0.00 |
| | No | 492 | 55 (11.20) | 437 (88.80) | |
| **Factors related to facilities** | | | | | |
| Type of hospital | Cancer centers | 1021 | 412 (40.40) | 609 (59.60) | 0.02 |
| | Other hospitals | 451 | 154 (34.10) | 297 (65.90) | |
| Bed capacity | Less than 200 | 165 | 75 (45.50) | 90 (54.50) | 0.05 |
| | 200 or more | 1307 | 491 (37.60) | 816 (62.40) | |

(*Continued*)

**Table 4.** (Continued)

| | | N | Satisfied n (%) | Not satisfied n (%) | P |
|---|---|---|---|---|---|
| Advertisements for CNSs and CNs | Yes | 1273 | 501 (39.40) | 772 (60.60) | 0.07 |
| | No | 199 | 65 (32.70) | 134 (67.30) | |
| Service system in the following four departments | Yes | 883 | 361 (40.90) | 522 (59.10) | 0.02 |
| | No | 589 | 205 (34.80) | 384 (65.20) | |
| Additional medical services• Palliative care practice | Yes | 593 | 236 (39.80) | 357 (60.20) | 0.38 |
| | No | 879 | 330 (37.50) | 579 (62.50) | |
| • Cancer patient counseling | Yes | 942 | 394 (41.80) | 548 (58.20) | 0.00 |
| | No | 530 | 172 (32.50) | 358 (67.50) | |
| • Outpatient palliative care management | Yes | 340 | 139 (40.90) | 201 (59.10) | 0.29 |
| | No | 1132 | 427 (37.70) | 705 (62.30) | |
| • Cancer pain palliation instruction | Yes | 830 | 349 (42.00) | 481 (58.00) | 0.00 |
| | No | 642 | 217 (33.80) | 425 (66.20) | |
| Requirements • Second opinion | Received | 1021 | 476 (46.60) | 725 (53.40) | 0.05 |
| | Not received | 271 | 90 (33.20) | 181 (66.80) | |
| • Collaboration team and the primary care physician | Received | 605 | 278 (46.00) | 327 (54.00) | 0.00 |
| | Not received | 867 | 288 (33.20) | 579 (66.80) | |
| • Operation of cancer regional alliances path | Received | 670 | 291 (43.40) | 379 (56.60) | 0.00 |
| | Not received | 802 | 275 (34.30) | 527 (65.70) | |
| • Training of local healthcare professionals | Received | 996 | 424 (42.60) | 572 (57.40) | 0.00 |
| | Not received | 476 | 142 (29.80) | 334 (70.20) | |
| • Hospital cancer registry | Received | 973 | 404 (41.50) | 569 (58.50) | 0.00 |
| | Not received | 499 | 162 (32.50) | 337 (67.50) | |
| Positive evaluation of CNS development | Yes | 531 | 286 (53.90) | 245 (46.10) | 0.00 |
| | No | 941 | 280 (29.80) | 661 (70.20) | |
| Positive evaluation of CN development | Yes | 898 | 451 (50.20) | 447 (49.80) | 0.00 |
| | No | 574 | 115 (27.00) | 459 (73.00) | |
| Appropriate staff allocation | Yes | 604 | 319 (52.81) | 285 (47.19) | 0.00 |
| | No | 868 | 247 (28.46) | 621 (71.54) | |
| **Factors related to the cancer care team** | | | | | |
| Coordination among healthcare professionals | Yes | 1384 | 548 (39.60) | 836 (60.40) | 0.00 |
| | No | 88 | 18 (20.50) | 70 (79.50) | |
| Independent activities | Yes | 1090 | 542 (49.70) | 548 (50.30) | 0.00 |
| | No | 382 | 24 (6.30) | 358 (93.70) | |
| Availability of conferences | Yes | 1235 | 515 (41.70) | 720 (58.30) | 0.00 |
| | No | 237 | 51 (21.50) | 186 (78.50) | |
| Job Type• Pharmacist | Involvement | 1350 | 528 (39.10) | 822 (60.90) | 0.44 |
| | Non-involvement | 122 | 38 (31.10) | 84 (68.90) | |
| • Medical social worker | Involvement | 1137 | 461 (40.50) | 676 (59.50) | 0.00 |
| | Non-involvement | 335 | 105 (31.30) | 230 (68.70) | |
| • Nutritionist | Involvement | 987 | 398 (40.30) | 589 (59.70) | 0.35 |
| | Non-involvement | 485 | 168 (34.60) | 317 (65.40) | |
| • Physical therapist | Involvement | 831 | 325 (39.10) | 506 (60.90) | 0.55 |
| | Non-involvement | 641 | 241 (37.60) | 400 (62.40) | |
| • Occupational therapist | Involvement | 547 | 217 (39.70) | 330 (60.30) | 0.46 |
| | Non-involvement | 925 | 349 (37.70) | 576 (62.30) | |
| • Speech therapist | Involvement | 307 | 119 (38.80) | 188 (61.20) | 0.90 |
| | Non-involvement | 1165 | 447 (38.40) | 718 (61.60) | |

(*Continued*)

**Table 4.** (Continued)

| | | | N | Satisfied n (%) | Not satisfied n (%) | P |
|---|---|---|---|---|---|---|
| | • Clinical psychologist | Involvement | 537 | 233 (43.40) | 304 (56.60) | 0.00 |
| | | Non-involvement | 935 | 333 (35.60) | 602 (64.40) | |

The threshold for years of experience was set at 5 years, because according to [32], 44.00% of CNs are hesitant to renew their initial 5-year contracts. Reasons given for their hesitation Include: "There is no satisfaction in the work" and they have "not been able to fully refresh."

performance in the role has been noticed have been correlated with increased job satisfaction [36]. Affective commitment is associated with increased job satisfaction and attachment to the organization [37, 38]. The results here suggest that it would be beneficial to provide CNSs and CNs with opportunities to maximize their abilities, which would enhance their affective commitment and job satisfaction.

CNs reported the presence of job satisfaction when staffing was appropriate in the department and when opportunities were available to perform cross-departmental and independent activities. However, job satisfaction was lower among CNs working on a ward, because doing so is associated with a higher level of exhaustion, which reduces job efficacy. Consistent with our findings, the appropriate staff allocation in wards or departments has been associated with a higher level of job satisfaction among CNs [17]. Therefore, flexible staffing is considered a guarantee that CNs will complete their activities. Job satisfaction increased at least five years after obtaining a CN qualification and when the individual received good evaluations from superiors. In addition, length of experience has been cited as a factor influencing the career development of CNs, and specialty careers with greater autonomy are associated with higher

**Table 5.** Logistic regression analysis.

| | | | OR | 95% CI | | | p |
|---|---|---|---|---|---|---|---|
| CNS | | | | | | | |
| Factors related to | | | | | | | |
| | Activities | Belongs to palliative care team | 2.64 | 1.08 | – | 6.45 | 0.04 |
| | | Presence of cross-departmental activities | 7.06 | 1.95 | – | 25.54 | 0.00 |
| | | Positive evaluation from senior staff | 13.15 | 3.19 | – | 54.19 | 0.00 |
| | | Presence of CN in radiation therapy nursing in the same institution | 2.91 | 1.08 | – | 7.84 | 0.03 |
| | Facilities | Positive evaluation of CNS development | 7.35 | 2.96 | – | 18.29 | 0.00 |
| | | Medical service fees: Cancer pain palliation instruction charges | 3.78 | 1.45 | – | 9.85 | 0.01 |
| | Team | Opportunity for independent activities | 11.34 | 2.04 | – | 62.99 | 0.01 |
| CN | | | | | | | |
| Factors related to | | | | | | | |
| | Activities | Belongs to ward department | 0.49 | 0.38 | – | 0.66 | 0.00 |
| | | Opportunities for cross-departmental activities | 2.24 | 1.57 | – | 3.19 | 0.00 |
| | | Positive evaluation from senior staff | 4.88 | 3.46 | – | 6.46 | 0.00 |
| | | Appropriate staff allocation | 1.75 | 1.35 | – | 2.27 | 0.00 |
| | | CN experience of more than 5 years | 1.91 | 1.44 | – | 2.54 | 0.00 |
| | Facilities | Bed capacity | 0.33 | 0.20 | – | 0.54 | 0.04 |
| | | Positive evaluation of CNS development | 1.37 | 1.01 | – | 1.86 | 0.04 |
| | | Positive evaluation of CN development | 2.13 | 1.55 | – | 2.93 | 0.00 |
| | Team | Opportunity for independent activities | 6.83 | 4.28 | – | 10.91 | 0.00 |

Adjusted for years of experience after acquiring a nursing license. Only variables that remained significant in the regression models are shown in this table.

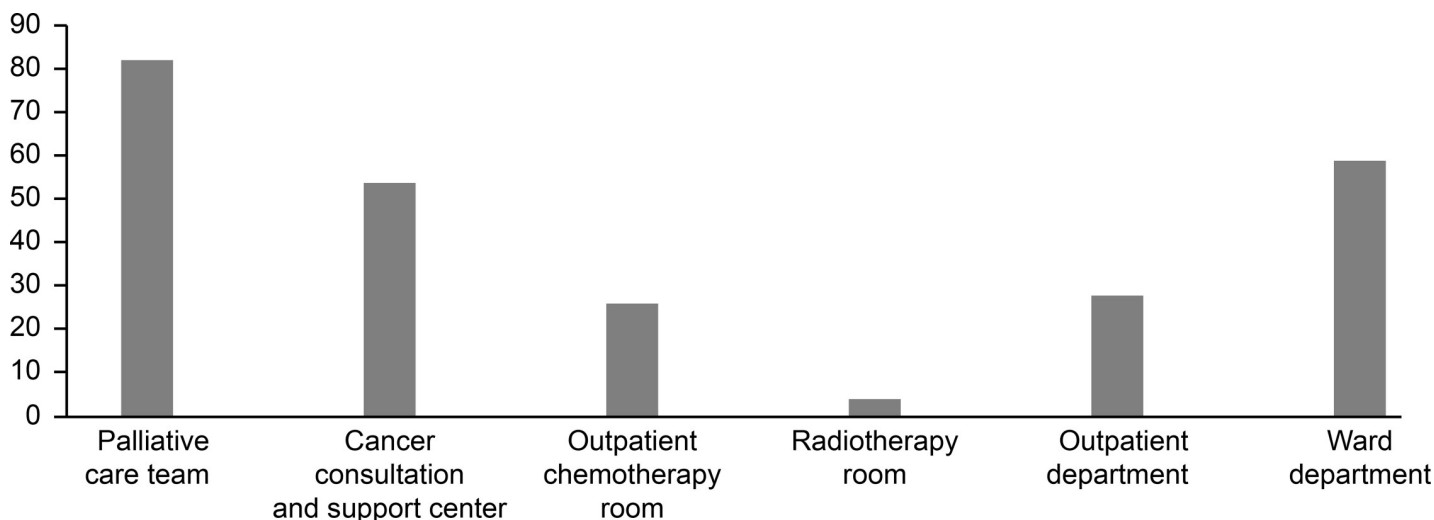

**Fig 1. Total number of CNSs in each affiliation.**

job satisfaction [39, 40]. For CNs, job satisfaction increased after the fifth year, when they can be fully active and solve problems independently.

In addition, job satisfaction was higher when the development of CNSs and CNs was conducted more proactively. Those who received positive evaluations from their senior staff tended to report the presence of job satisfaction. Because work engagement is increased by the availability of resources such as support from senior staff and colleagues [41], and lack of social support from senior staff is associated with the intention to quit a job [42], a high rating and support from senior staff is thought to increase the job satisfaction of CNSs and CNs. Thus, for CNSs and CNs to develop their professional abilities [43], receiving a favorable evaluation from a supervisor is essential.

Although previous studies found the presence of job satisfaction among CNSs and CNs who are more active in teaching, the absence of supportive colleagues reportedly influence job

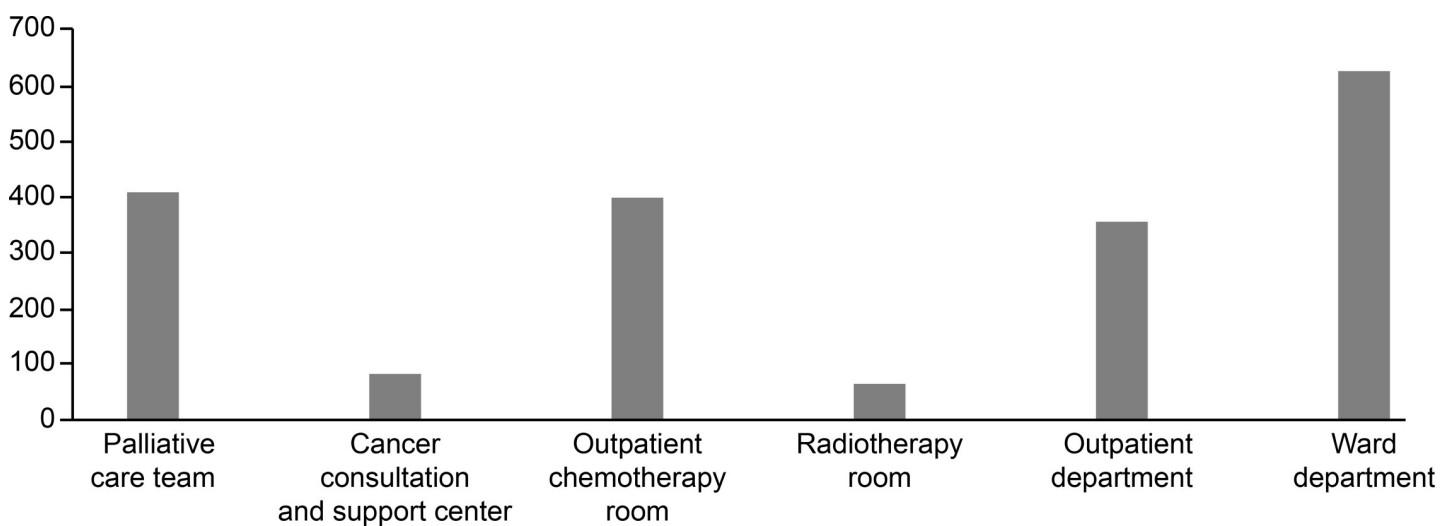

**Fig 2. Total number of CNs in each affiliation.**

satisfaction as well [41–43]. Moreover, affective commitment, which involves equating one's personal values with those of the organization, is strongly associated with job motivation in nurses [44] and enhances job satisfaction. Importantly, the perception of support from colleagues is related to this affective commitment [45, 46]. For example, perceptions of mutual support between CNSs and CNs seemed to enhance job satisfaction.

Of the factors related to facilities, a positive evaluation of the development of CNs and CNSs was associated with job satisfaction in both CNs and CNSs.

CNSs' job satisfaction was significantly higher when institutions charged additional fees for pain relief management. The fee for opioid use in Japan is approximately 5.00% of that in Western countries [47]. However, it is difficult to determine whether appropriate pain care is provided in Japan. In this context, there is often misunderstanding due to the lack of knowledge about opioids of both patients and healthcare professionals [48]. Therefore, the reevaluated and newly established system for medical service fees that provides instructions on pain palliation in cancer treatment may offer a sense of fulfillment and satisfaction to CNSs, because they are able to provide care that alleviates pain and improves their patients' quality of life.

Moreover, job satisfaction was higher among CNs who worked at an institution with a capacity of less than 200 beds. Reportedly, professional autonomy is affected by the size and management system of the organization, not by individual traits [40]. The opinions of nurses are addressed by administrators more readily in smaller institutions than in larger ones, and they can demonstrate their competency more easily, which may explain the association between bed capacity and job satisfaction. Of the factors related to the cancer care team, opportunity for independent activities was associated with the job satisfaction of both CNs and CNSs. To experience job satisfaction, individuals with an internal locus of control need an environment in which they can effectively use their roles to determine the outcomes of their actions [35]. Hence, for CNSs and CNs to provide care as intended, an environment is required in which they can fully use their cultivated knowledge, skills, and designated roles. This tendency toward an internal locus of control might have increased job satisfaction in the current study.

Certified nursing was an outcome of the notion that nurses should collaborate and engage in cross-sectional activities beyond their wards. Engaging in such activities likely increases collaboration and the job satisfaction of CNSs. It is important that organizations adopt practices based on the findings of this study. Moreover, the working conditions and job satisfaction of CNSs and CNs reported in this study are closely connected with policy. Therefore, our findings should be used to guide policy in Japan in the future.

Several previous studies have been conducted on the job satisfaction of nurses in Japanese institutions. They highlight the complex nature of factors influencing job satisfaction and importance thereof in staff retention [49–55].

Our study has several limitations. Because fewer responses were obtained from CNSs (only 200 of the total 483 CNSs) than CNs (1,472 responses), the 95% confidence interval of the logistic regression was rather large. In addition, the effects of age and sex were not considered. The group sizes of participants working in radiotherapy or wards with lower bed capacity were skewed. Finally, because of the explorative character of our study, we did not correct for multiple testing. Therefore, these results must be generalized with caution.

## Conclusion

Our study identified the factors related to activities, facilities, and the cancer care team associated with the job satisfaction of CNSs and CNs, and highlighted potential avenues through which to enhance job satisfaction through the Basic Plan to Promote Cancer Control in Japan.

Whereas the majority of the factors identified were shared by CNs and CNSs, several were unique to one of the groups. Moreover, the findings have implications for hospital administrators aiming to retain staff who might otherwise be hesitant to stay because of job dissatisfaction. Finally, changes regarding the CN certification may be beneficial.

## Supporting information

**S1 Data.**
(XLSX)

**S2 Data.**
(XLSX)

## Acknowledgments

Editorial support in the form of medical writing, assembling tables, creating high-resolution images based on authors' detailed directions, collating author comments, copyediting, fact-checking, and referencing was provided by Editage, Cactus Communications.

## Author Contributions

**Conceptualization:** Hidenori Arai.

**Data curation:** Masaki Kitajima, Chiharu Miyata.

**Formal analysis:** Masaki Kitajima, Chiharu Miyata, Hidenori Arai.

**Funding acquisition:** Masaki Kitajima, Hidenori Arai.

**Investigation:** Masaki Kitajima, Chiharu Miyata.

**Methodology:** Masaki Kitajima, Chiharu Miyata, Hidenori Arai.

**Supervision:** Keiko Tamura, Ayae Kinoshita, Hidenori Arai.

**Writing – original draft:** Masaki Kitajima.

**Writing – review & editing:** Chiharu Miyata, Ayae Kinoshita, Hidenori Arai.

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
