## [Decision Letter · Decision Letter 0]

18 Dec 2019

PONE-D-19-28473

Job satisfaction of certified nurse specialists and certified nurses in cancer care centers in Japan: A large cross-sectional study

PLOS ONE

Dear Professor Arai,

Thank you for submitting your manuscript to PLOS ONE. After careful consideration, we feel that it has merit but does not fully meet PLOS ONE’s publication criteria as it currently stands. Therefore, we invite you to submit a revised version of the manuscript that addresses the points raised during the review process.

We would appreciate receiving your revised manuscript by Feb 01 2020 11:59PM. To enhance the reproducibility of your results, we recommend that if applicable you deposit your laboratory protocols in protocols.io, where a protocol can be assigned its own identifier (DOI) such that it can be cited independently in the future. For instructions see: http://journals.plos.org/plosone/s/submission-guidelines#loc-laboratory-protocols

We look forward to receiving your revised manuscript.

Kind regards,

Janhavi Ajit Vaingankar

Academic Editor

PLOS ONE

Journal Requirements:

3. Thank you for including your ethics statement; "The study protocol was approved by the Institutional Ethics Committee with protocol number E2072. The purpose of the study, benefits of participation, confirmation of voluntary participation, and an assurance that all data obtained in the present study would be used only for scientific purposes and anonymously were communicated to all participants in written form. All participants gave informed consent. To protect confidentiality, personal data were kept separately from the completed questionnaires, which were coded."

Reviewers' comments:

Reviewer's Responses to Questions

**Comments to the Author**

1. Is the manuscript technically sound, and do the data support the conclusions?

Reviewer #1: No

Reviewer #2: Yes

2. Has the statistical analysis been performed appropriately and rigorously? 

Reviewer #1: No

Reviewer #2: Yes

3. Have the authors made all data underlying the findings in their manuscript fully available?

Reviewer #1: No

Reviewer #2: No

4. Is the manuscript presented in an intelligible fashion and written in standard English?

Reviewer #1: Yes

Reviewer #2: Yes

5. Review Comments to the Author

Reviewer #1: The manuscript touches an important topic by trying to assess Job satisfaction of certified nurse specialists and certified nurses in cancer care centers in Japan. However, I have some doubts how this study was conducted:

Title:

- Your title of this study is “to assess Job satisfaction of certified nurse specialists and certified nurses in cancer care centers in Japan”. However; there is no finding in your study which shows the level of job satisfaction among these two group of nurses. Therefore, you have to either present this finding or modify your title.

Abstract:

- In the background part of abstract, information about nurses’ job satisfaction such as level of job satisfaction and burden of job dissatisfaction on patients should be presented from the available literature.

- I am not clear with your sample size calculation and sampling procedure. How 200 certified nurse specialist and 1,472 certified nurses were recruited and participated in the study?

- The authors should indicate the level of job satisfaction of both certified nurse specialist and certified nurses

- Indicate all variables associated with job satisfaction separately for certified nurse specialist and certified nurses, and mention their respective AOR with CI.

- The authors should present the conclusion of their finding in the conclusion section of the abstract

Introduction:

- Based on the available literature, the authors should present the level of job satisfaction among nurses in global, regional and local perspectives

- The authors should also show what the literature says about the burden/impact of nurses’ job dissatisfaction on cancer patients. The authors should also mention important factors influencing job satisfaction based on the available literature on this particular topic

- I recommend the authors to remove table 1 from introductory part of this manuscript, because information in the table are already mentioned in narration

- The statement “No study in the study area” don’t justify the need of your study. The authors should show gap of previous studies (global perspectives) and justify the need of your studies

Methods:

- I am not sure whether the level job satisfaction of nurses was measured in this study. If so, which tool was used to measure this job satisfaction? Was this tools validity and reliability checked?

- Why you conducted pre-test on only 4 nurses? As a rule, on how much percentage of sample size pre-test should be conducted?

- How you determined the sample size? What was your sampling procedure? The authors should describe the mechanism how participants were recruited and participated in the study

- Since the data were collected through mail, I doubt how the informed consent was obtained.

- My major issue with this research is how job satisfaction was assessed/measured and categorized as “satisfaction” and “dissatisfaction”. Therefore, the authors should clearly explain this in the methods

- Your method of analysis is not clear. Please, explain your method of analysis in detail

Results:

- Your objective is to assess Job satisfaction of certified nurse specialists and certified nurses in cancer care centers in Japan. Where is the result which indicate level of job satisfaction for certified nurse specialist and certified nurses? What percentages of certified nurse specialists and certified nurses were satisfied and dissatisfied with their job?

Discussion:

- The authors should re-write the discussion. The authors only stated and discussed their finding and didn’t compare their finding with other study findings. The authors should discuss only the key finding of their study with previous study findings.

Conclusion:

- The authors should write the conclusion of the study finding separately

Reviewer #2: Thank you for the opportunity to review this paper. This paper is novel in that it is the first to evaluate the factors associated with job satisfaction among nurses involved in cancer care in Japan, which is an important group of healthcare professionals.

Introduction

1) In Table 1, years of clinical experience to qualify for CNSs for university graduates is stated as “0” years. Please confirm this is correct.

2) Reference 22 appears to be out of context w.r.t addressing job satisfaction among nurses. Please confirm if the reference is appropriate. (line 91)

Methods

1) The interviews conducted (line 111) needs to be further clarified. E.g., with who?

2) “a logistic regression analysis was conducted with satisfaction or dissatisfaction as the dependent variable” (line 183)- please provide more information to explain how the dependent variable is being determined. Given the proposed title of the manuscript, it is also important to report on job satisfaction status as descriptive statistics in Table 3.

3) “years of experience as the control variable” (line 185) refers to years of experience after acquiring nursing license or after acquiring CN qualification?

Results:

Overall: The regression analysis for CNSs (n=200) is under-powered given the large no. of independent variables involved (which i suppose to be 20 of them).

1) “remaining participants worked at other hospitals (e.g., general hospitals or clinics)” (line 196-197). It is unclear if these participants were also involved in cancer care? This appears to contradicts the statement under section-Participants which stated that “A total of 1,696 nurses working in cancer centers and hospitals responded to the mailed questionnaire. (line 160-161)”. Please clarify.

2) “51.6% of the nurses had a position title (e.g., head or chief nurse), whereas the others were staff nurses” (line 203), is this established based on “position” in Table 3 where managers are “head or chief nurse”? If so, the reported % does not seem to be correct.

3) I noticed significant differences in 5 (not 3) items with cancer team care for Table 5. Please correct text under results accordingly (line 208 and line 211).

4) “The results of the χ2 test were used in a logistic regression analysis to identify variables particularly important to job satisfaction among CNSs and CNs.” (line 228-229). This needs to be highlighted in the methodology and it is unclear how results from chi square tests were used to identify variables for inclusion in logistic regressions?

Was it that only items with p<0.05 at univariate analysis were selected as explanatory variables in the logistic regressions? If so, why is “presence of CN in radiation therapy nursing” included and remained significant in the regression analysis for CNS (Table 6) when it was reported as “n.s” in the univariate analysis for CNS. Please clarify. I have concern on the over-loading of the model, particularly given the small sample in CNS analysis. Is correction for multiple testing considered?

5) Table 6 shows only variables that remain significant in regression models? If so, please state this as legend in the Table.

6) “when a CN was involved in radiation therapy nursing within the same institution” (line 242-243); please clarify if this is a factor related to activity or facilities? It is reported as a factor for facilities.

7) Suggest to use the terms “presence” or “absence” of job satisfaction instead of “higher” or “lower” level of job satisfaction when reporting the results.

Discussion

1) “a high number of respondents (1,404) were affiliated with a single ward” (line 294). Figure 2 shows that less than 700 (based on bar height) were involved in ward department. Please clarify.

6. PLOS authors have the option to publish the peer review history of their article (what does this mean?). If published, this will include your full peer review and any attached files.

Reviewer #1: No

Reviewer #2: No

---

## [Author Response · Author response to Decision Letter 0]

27 Jan 2020

Response to the Editor and Reviewers

Editor:

When submitting your revision, we need you to address these additional requirements. Please ensure that your manuscript meets PLOS ONE's style requirements, including those for file naming.

Response: We have renamed the files in accordance with the journal requirements and rechecked the text for compliance with the formatting requirements.

We note that you have stated that you will provide repository information for your data at acceptance. Should your manuscript be accepted for publication, we will hold it until you provide the relevant accession numbers or DOIs necessary to access your data. If you wish to make changes to your Data Availability statement, please describe these changes in your cover letter and we will update your Data Availability statement to reflect the information you provide.

Response: We confirm that we will make the manuscript data publicly available upon acceptance of the article and agree with the outlined procedure.

Please amend your current ethics statement to include the full name of the ethics committee/institutional review board(s) that approved your specific study. Once you have amended this/these statement(s) in the Methods section of the manuscript, please add the same text to the “Ethics Statement” field of the submission form (via “Edit Submission”). For additional information about PLOS ONE ethical requirements for human subjects research, please refer to http://journals.plos.org/plosone/s/submission-guidelines#loc-human-subjects-research.

Response: We have specified the name of the Ethics Committee (“Institutional Ethics Committee of the Kyoto University Graduate School of Medicine and Faculty of Medicine Hospital”) in the Methods section and the ethical statement.

PLOS requires an ORCID iD for the corresponding author in Editorial Manager on papers submitted after December 6th, 2016. Please ensure that you have an ORCID iD and that it is validated in Editorial Manager. To do this, go to ‘Update my Information’ (in the upper left-hand corner of the main menu), and click on the Fetch/Validate link next to the ORCID field. This will take you to the ORCID site and allow you to create a new iD or authenticate a pre-existing iD in Editorial Manager. Please see the following video for instructions on linking an ORCID iD to your Editorial Manager account: https://www.youtube.com/watch?v=_xcclfuvtxQ.

Response: We have provided the ORCiD ID for the corresponding author as requested.

Reviewer #1: 

The manuscript touches an important topic by trying to assess Job satisfaction of certified nurse specialists and certified nurses in cancer care centers in Japan. However, I have some doubts how this study was conducted:

Title:

- Your title of this study is “to assess Job satisfaction of certified nurse specialists and certified nurses in cancer care centers in Japan”. However; there is no finding in your study which shows the level of job satisfaction among these two group of nurses. Therefore, you have to either present this finding or modify your title.

Response: We thank the reviewer for this helpful comment. Job satisfaction was present in 38.45% of CNs and 49% of CNSs, but absent in 61.55% of CNs and 51% of CNSs. We have added this information to the revised manuscript (page 12, lines 202-203 and Table 2). Furthermore, to more exactly convey the topic and findings of the paper, we have modified the title to: “Factors affecting the job satisfaction of certified nurse specialists and certified nurses in cancer care in Japan: Analysis based on the Basic Plan to Promote Cancer Control Programs.”

Abstract:

- In the background part of abstract, information about nurses’ job satisfaction such as level of job satisfaction and burden of job dissatisfaction on patients should be presented from the available literature.

Response: We have added a statement on the level of job satisfaction and its consequences to the background part of the abstract. Due to the 300 words limit for the abstract, we had to keep the statement concise (page 2, lines 21-23).

-I am not clear with your sample size calculation and sampling procedure. How 200 certified nurse specialist and 1,472 certified nurses were recruited and participated in the study?

Response: The sample size was restricted by the number of eligible qualified and recruited CNs and CNSs. Overall, 2,378 eligible CNs were identified and contacted; 1,486 replied, and 1,472 among them provided valid answers. Overall, 514 eligible CNSs were identified and contacted; 210 replied, and 200 among them provided valid answers.

-The authors should indicate the level of job satisfaction of both certified nurse specialist and certified nurses

Response: Job satisfaction was present in 38.45% of CNs and 49% of CNSs but was absent in 61.55% of CNs and 51% of CNSs. We have added this information to the manuscript (page 12, lines 202-203 and Table 2).

-Indicate all variables associated with job satisfaction separately for certified nurse specialist and certified nurses, and mention their respective AOR with CI.

Response: All variables associated with job satisfaction were listed separately, and their odds ratios and p-values were specified. Unfortunately, the word limit for the abstract (300 words) did not allow listing also the 95% CI in the abstract; however, this information is described in the main text.

-The authors should present the conclusion of their finding in the conclusion section of the abstract

Response: We have stated the conclusion of the study in the conclusion section of the abstract (page 3, lines 45-48).

Introduction:

- Based on the available literature, the authors should present the level of job satisfaction among nurses in global, regional and local perspectives

Response: The introduction has been modified (page 5, lines 86-91).

- The authors should also show what the literature says about the burden/impact of nurses’ job dissatisfaction on cancer patients. The authors should also mention important factors influencing job satisfaction based on the available literature on this particular topic

Response: We have mentioned important factors affecting job satisfaction in the introduction (page 5, lines 86-91). Unfortunately, the effect of nurse job dissatisfaction on cancer patient outcomes has not been investigated well. We have added that as a future research direction in the discussion.

- I recommend the authors to remove table 1 from introductory part of this manuscript, because information in the table are already mentioned in narration

Response: We have removed Table 1 from the revised version of the manuscript.

- The statement “No study in the study area” don’t justify the need of your study. The authors should show gap of previous studies (global perspectives) and justify the need of your studies

Response: We have modified the justification for the study (page 6, lines 98-102).

Methods:

- I am not sure whether the level job satisfaction of nurses was measured in this study. If so, which tool was used to measure this job satisfaction? Was this tools validity and reliability checked?

Response: To measure job satisfaction, we asked the question, "Are you satisfied with your job?” Originally, we used a scale with three levels, “satisfied,” “somewhat satisfied,” and “dissatisfied.” However, upon review, we combined the first two levels to yield a two-level scale, “satisfied” and ”dissatisfied.” We have added this information to the text of the manuscript (page 8, lines 142-145).

- Why you conducted pre-test on only 4 nurses? As a rule, on how much percentage of sample size pre-test should be conducted?

Response: Only 4 nurses were included in the pre-test due to organizational and technical reasons despite our intention to conduct a more extensive pre-test. Since the pre-test was conducted in person and was time-consuming, it was difficult to recruit nurses willing to commit the time for the pre-test. Furthermore, the nurses we managed to recruit were at a geographical distance from us, which necessitated travel to their institutions, further increasing the time load.

- How you determined the sample size? What was your sampling procedure? The authors should describe the mechanism how participants were recruited and participated in the study

Response: As mentioned earlier in our response, we contacted all eligible qualified CNs and CNSs. Overall, there were 2,378 eligible CNs; 1,486 replied, and 1,472 among them provided valid answers. Overall, there were also 514 eligible CNSs; 210 replied, and 200 among them provided valid answers.

- Since the data were collected through mail, I doubt how the informed consent was obtained.

Response: The contacted CNs and CNSs had the opportunity to select also the item “I do not agree to participate.” in the questionnaire we sent them.

- My major issue with this research is how job satisfaction was assessed/measured and categorized as “satisfaction” and “dissatisfaction”. Therefore, the authors should clearly explain this in the methods

Response: To measure job satisfaction, we asked the question, "Are you satisfied with your job?” Originally, we used a scale with three levels, “satisfied,” “somewhat satisfied,” and “dissatisfied.” However, upon review, we combined the first two levels to yield a two-level scale, “satisfied” and ”dissatisfied.”

- Your method of analysis is not clear. Please, explain your method of analysis in detail

Response: The description of the data analysis has been revised in the text.

Results:

- Your objective is to assess Job satisfaction of certified nurse specialists and certified nurses in cancer care centers in Japan. Where is the result which indicate level of job satisfaction for certified nurse specialist and certified nurses? What percentages of certified nurse specialists and certified nurses were satisfied and dissatisfied with their job?

Response: Job satisfaction was present in 38.45% of CNs and 49% of CNSs, and absent in 61.55% of CNs and 51% of CNSs (page 12, lines 202-203 and Table 2). 

Discussion:

- The authors should re-write the discussion. The authors only stated and discussed their finding and didn’t compare their finding with other study findings. The authors should discuss only the key finding of their study with previous study findings.

Response: We have modified the discussion to focus more on the comparison of our findings with the results of previous investigations.

Conclusion:

- The authors should write the conclusion of the study finding separately

Response: The Conclusion has been separated into an individual section (page 27, lines 390-396). 

Reviewer #2: Thank you for the opportunity to review this paper. This paper is novel in that it is the first to evaluate the factors associated with job satisfaction among nurses involved in cancer care in Japan, which is an important group of healthcare professionals.

Introduction

1) In Table 1, years of clinical experience to qualify for CNSs for university graduates is stated as “0” years. Please confirm this is correct.

Response: We thank the Reviewer for this comment. The second Reviewer of the manuscript requested that we remove Table 1 due to the presentation of the data in the text. Therefore, Table 1 has been removed from the revised version of the manuscript.

2) Reference 22 appears to be out of context w.r.t addressing job satisfaction among nurses. Please confirm if the reference is appropriate. (line 91)

Response: We thank the Reviewer for pointing out this inconsistency. We have removed reference 22 from the revised manuscript.

Methods

1) The interviews conducted (line 111) needs to be further clarified. E.g., with who?

Response: We meant the pre-test interviews with CNs. To avoid confusion, this phrase was removed from the revised version of the manuscript.

2) “a logistic regression analysis was conducted with satisfaction or dissatisfaction as the dependent variable” (line 183)- please provide more information to explain how the dependent variable is being determined. Given the proposed title of the manuscript, it is also important to report on job satisfaction status as descriptive statistics in Table 3.

Response: We have reported on job satisfaction in the descriptive statistics Table, which is currently Table 2 in the revised manuscript. To measure job satisfaction, we asked the question, "Are you satisfied with your job?” Originally, we used a scale with three levels, “satisfied,” “somewhat satisfied,” and “dissatisfied.” However, upon review, we combined the first two levels to yield a two-level scale, “satisfied” and ”dissatisfied.” (page 12, lines 202-203 and Table 2).

3) “years of experience as the control variable” (line 185) refers to years of experience after acquiring nursing license or after acquiring CN qualification?

Response: "Years of experience after acquiring nursing license" was used as a control variable, whereas "years of experience after acquiring CNS or CN qualification" was utilized as an independent variable. We have clarified this information in the text.

Results:

Overall: The regression analysis for CNSs (n=200) is under-powered given the large no. of independent variables involved (which i suppose to be 20 of them)

Response: We contacted all eligible qualified CNs and CNSs. Overall, there were 514 eligible CNSs; 210 replied, and 200 among them provided valid answers. Thus, the CNS sample size was limited by the number of individuals eligible and willing to participate. We have stated this fact as a limitation of our study.

1) “remaining participants worked at other hospitals (e.g., general hospitals or clinics)” (line 196-197). It is unclear if these participants were also involved in cancer care? This appears to contradicts the statement under section-Participants which stated that “A total of 1,696 nurses working in cancer centers and hospitals responded to the mailed questionnaire. (line 160-161)”. Please clarify.

Response: We apologize for this unclarity. These participants worked in general hospitals or clinics, but they were also involved in cancer care. We have modified both statements, to which the Reviewer referred, to clarify this point (page 12, lines 204-206).

2) “51.6% of the nurses had a position title (e.g., head or chief nurse), whereas the others were staff nurses” (line 203), is this established based on “position” in Table 3 where managers are “head or chief nurse”? If so, the reported % does not seem to be correct.

Response: We apologize for this inconsistency and have changed the value to the correct value of 60.5%. 

3) I noticed significant differences in 5 (not 3) items with cancer team care for Table 5. Please correct text under results accordingly (line 208 and line 211).

Response: We thank the Reviewer for pointing out this inconsistency; we have corrected it (page 14, lines 219-220).

4) “The results of the χ2 test were used in a logistic regression analysis to identify variables particularly important to job satisfaction among CNSs and CNs.” (line 228-229). This needs to be highlighted in the methodology and it is unclear how results from chi square tests were used to identify variables for inclusion in logistic regressions? Was it that only items with p<0.05 at univariate analysis were selected as explanatory variables in the logistic regressions? If so, why is “presence of CN in radiation therapy nursing” included and remained significant in the regression analysis for CNS (Table 6) when it was reported as “n.s” in the univariate analysis for CNS. Please clarify. I have concern on the over-loading of the model, particularly given the small sample in CNS analysis. Is correction for multiple testing considered?

Response: We thank the Reviewer for pointing out this unclarity. We have specified in the revised manuscript that only items with p < 0.05 in the univariate analysis were selected as explanatory variables in the logistic regressions (page 11, lines 186-188). The presence of CN in radiation therapy nursing “n.s” report in the univariate analysis for CNS was a mistake that we have corrected. Due to the exploratory character of our study, we did not correct for multiple testing. We have stated this fact as a limitation in the revised version of the manuscript (page 27, lines 387-388).

5) Table 6 shows only variables that remain significant in regression models? If so, please state this as legend in the Table.

Response: We have stated in the note to the table in the revised manuscript that only variables that remain significant in regression models are shown in Table 6.

6) “when a CN was involved in radiation therapy nursing within the same institution” (line 242-243); please clarify if this is a factor related to activity or facilities? It is reported as a factor for facilities.

Response: We thank the Reviewer for pointing out this inconsistency and apologize for it. “When a CN was involved in radiation therapy nursing within the same institution” was a factor associated with facilities. We have corrected the text accordingly.

7) Suggest to use the terms “presence” or “absence” of job satisfaction instead of “higher” or “lower” level of job satisfaction when reporting the results.

Response: We have replaced the phrases “higher” or “lower” level of job satisfaction with “presence” or “absence” of job satisfaction.

Discussion

1) “a high number of respondents (1,404) were affiliated with a single ward” (line 294). Figure 2 shows that less than 700 (based on bar height) were involved in ward department. Please clarify.

Response: We thank the reviewer for pointing out this inconsistency. By “affiliated with a single ward” we meant “affiliated with a single department,” including a ward or outpatient department, as well as a chemotherapy or radiotherapy room. We have modified the phrase in the text correspondingly (page 23, lines 305-306).

---

## [Decision Letter · Decision Letter 1]

25 Feb 2020

PONE-D-19-28473R1

Factors affecting job satisfaction of certified nurse specialists and certified nurses in cancer care in Japan: Analysis based on the Basic Plan to Promote Cancer Control Programs

PLOS ONE

Dear Professor Arai,

Thank you for submitting your manuscript to PLOS ONE. After careful consideration, we feel that it has merit but does not fully meet PLOS ONE’s publication criteria as it currently stands. Therefore, we invite you to submit a revised version of the manuscript that addresses the points raised during the review process.

ACADEMIC EDITOR: 

Please consider the comments by the third reviewer pertaining to data analysis. You may wish to revise your analysis or justify your analytical approach in the light of these comments.

We would appreciate receiving your revised manuscript by Apr 10 2020 11:59PM. To enhance the reproducibility of your results, we recommend that if applicable you deposit your laboratory protocols in protocols.io, where a protocol can be assigned its own identifier (DOI) such that it can be cited independently in the future. For instructions see: http://journals.plos.org/plosone/s/submission-guidelines#loc-laboratory-protocols

We look forward to receiving your revised manuscript.

Kind regards,

Janhavi Ajit Vaingankar

Academic Editor

PLOS ONE

Reviewers' comments:

Reviewer's Responses to Questions

**Comments to the Author**

1. If the authors have adequately addressed your comments raised in a previous round of review and you feel that this manuscript is now acceptable for publication, you may indicate that here to bypass the “Comments to the Author” section, enter your conflict of interest statement in the “Confidential to Editor” section, and submit your "Accept" recommendation.

Reviewer #2: (No Response)

Reviewer #3: (No Response)

2. Is the manuscript technically sound, and do the data support the conclusions?

Reviewer #2: Yes

Reviewer #3: No

3. Has the statistical analysis been performed appropriately and rigorously? 

Reviewer #2: Yes

Reviewer #3: No

4. Have the authors made all data underlying the findings in their manuscript fully available?

Reviewer #2: Yes

Reviewer #3: No

5. Is the manuscript presented in an intelligible fashion and written in standard English?

Reviewer #2: Yes

Reviewer #3: No

6. Review Comments to the Author

Reviewer #2: The authors have adequately addressed most of comments except one:

1) "when a CN was involved in radiation therapy nursing within the same institution” (line 248-252 in the clean version);

please clarify if this is a factor related to activity or facilities? The author replied in his/her comment that this is a factor related to facilities which is confusing and it does not tally with information presented in the tables. I did not also see any change in the text.

Others:

1) Line 217 to 218. All reported numbers should be updated. 20->21, 30->32, 10->11

Reviewer #3: The use of the abbreviations is not consistent. For example, CN or CNs, CNSs. I would suggest to remove all acronyms from the abstract in the least and standardize their use throughout the manuscript. The use of the acronyms has to be grammatically correct i.e. used in singular or plural forms. Consider using ‘certified nurse or nurse specialists’ in the sentences. Please spell chi-square in full.

Since this is a cross-sectional study, refrain from using terms such as ‘predictive’. Change these to ‘association’ instead.

State all proportions with upto one decimal points and other estimates upto 2 or 3 (keep it consistent).

Please state full p values instead of asterisks in the tables.

I am unable to understand the significant of having the analysis based on the Basic Plan to Promote Cancer Control Programs. Perhaps in the introduction this should be described and its relevance to this study should be highlighted. It may also be necessary to state why it was of interest to study CN and CNS groups separately. Just stating there were no previous studies may not be enough.

The discussion is not informative and needs to be clearly focused on the main findings of the study and the distinctions between the two groups and their practice implications.

In terms of analysis, some of the groups are highly skewed eg. Proportions working in radiotherapy, wards with lower bed capacity. Years of clinical nurse experience also needs to be carefully looked at since this has been used as control variable in the logistic regression. Although authors justify using 5 year cut off, I believe this again would be very skewed population given the mean duration was 19 years in the sample. The univariate analysis need to relooked at carefully before presenting multivariable findings. On that note, the correct term would be ‘multivariable’ instead of ‘multivariate’

Overall, I found this article quite difficult to follow. Given that it is currently quite long, authors need to edit the content carefully after considering the above comments. This could be achieved by shortening the introduction, avoiding repetitions between the tables and text in the results and trimming the discussion. Parts of the manuscript also need better flow and focus. For example, why is the statement ‘The study adheres to the STROBE checklist’ included in the job satisfaction questionnaire section? The intent of mentioning ‘Basic Plan to Promote Cancer Control Programs’ is not clear. Analysis needs to be reviewed thoroughly and reanalyzed as necessary to be eligible for publication.

7. PLOS authors have the option to publish the peer review history of their article (what does this mean?). If published, this will include your full peer review and any attached files.

Reviewer #2: No

Reviewer #3: No

---

## [Author Response · Author response to Decision Letter 1]

31 Mar 2020

Response to the Editor and Reviewers

Academic Editor:

Please consider the comments by the third reviewer pertaining to data analysis. You may wish to revise your analysis or justify your analytical approach in the light of these comments.

Response: Thank you for the opportunity to revise our manuscript. We have conducted additional statistical analysis in accordance with the comments of Reviewer 3.

Reviewer #2: 

The authors have adequately addressed most of comments except one:

1) "when a CN was involved in radiation therapy nursing within the same institution” (line 248-252 in the clean version); please clarify if this is a factor related to activity or facilities? The author replied in his/her comment that this is a factor related to facilities which is confusing and it does not tally with information presented in the tables. I did not also see any change in the text.

Response: We apologize for this mistake. In the revised manuscript, we changed this to “when a CN was involved in radiation therapy nursing within the same institution” as a factor related to activity (page 21, lines 234-235).

Others:

1) Line 217 to 218. All reported numbers should be updated. 20->21, 30->32, 10->11

Response: Thank you for pointing out this inconsistency. We have updated the reported numbers accordingly (page 14, lines 205-208).

Reviewer #3: 

The use of the abbreviations is not consistent. For example, CN or CNs, CNSs. I would suggest to remove all acronyms from the abstract in the least and standardize their use throughout the manuscript. The use of the acronyms has to be grammatically correct i.e. used in singular or plural forms. Consider using ‘certified nurse or nurse specialists’ in the sentences. Please spell chi-square in full.

Response: Thank you for this comment. We have ensured that the use of abbreviations in the revised version of the manuscript is consistent by introducing the following modifications: (1) We have removed all acronyms from the abstract. (2) In the main text, we have used CN and CNS in singular form and CNs and CNSs in plural form. (3) In the abstract, we have used the expression “certified nurses or nurse specialists” as suggested. (4) We have spelled out “chi-square” in full.

Since this is a cross-sectional study, refrain from using terms such as ‘predictive’. Change these to ‘association’ instead.

Response: We deleted all terms indicating a causative effect such as “predictive” and changed them to terms indicating an “association” as requested.

State all proportions with upto one decimal points and other estimates upto 2 or 3 (keep it consistent).

Response: We have modified the text to consistently present estimates up to 2 decimal points.

Please state full p values instead of asterisks in the tables.

Response: We have provided the full p-values instead of asterisks in the tables.

I am unable to understand the significant of having the analysis based on the Basic Plan to Promote Cancer Control Programs. Perhaps in the introduction this should be described and its relevance to this study should be highlighted. It may also be necessary to state why it was of interest to study CN and CNS groups separately. Just stating there were no previous studies may not be enough.

Response: In the introduction, we now elaborate why we chose to base the analysis on the Basic Plan to Promote Cancer Control Programs (page 5, lines 75-77 of the clean copy). The roles of certified nurses and nurse specialists are regulated by the Basic Plan to Promote Cancer Control Programs. Therefore, it is important to consider this plan when assessing nurses’ job satisfaction. Furthermore, considering the plan would help us make more relevant suggestions for measures from the Japanese Ministry of Health, Labour and Welfare to improve job satisfaction. The roles of certified nurses and nurse specialists partially overlap, but also differ in several aspects. Therefore, it is important to analyze the factors influencing job satisfaction in these groups separately to identify overlapping and distinct factors. This also facilitates the identification of measures that can improve the job satisfaction of certified nurses and nurse specialists.

The discussion is not informative and needs to be clearly focused on the main findings of the study and the distinctions between the two groups and their practice implications.

Response: We have extensively revised the discussion accordingly.

In terms of analysis, some of the groups are highly skewed eg. Proportions working in radiotherapy, wards with lower bed capacity. Years of clinical nurse experience also needs to be carefully looked at since this has been used as control variable in the logistic regression. Although authors justify using 5 year cut off, I believe this again would be very skewed population given the mean duration was 19 years in the sample. The univariate analysis need to relooked at carefully before presenting multivariable findings. On that note, the correct term would be ‘multivariable’ instead of ‘multivariate’

Response: Thank you for your comment. In the revised manuscript, we conducted an univariate analysis on the association of job experience of “less than 19 years”/“19 years and above” with “job satisfaction” (page 14, Table 3). No significant association was found for both certified nurses and nurse specialists. In addition, we replaced the term “multivariate” with “multivariable.” We have stated in the limitations statement that some groups such as participants working in radiotherapy or wards with lower bed capacity were skewed.

Overall, I found this article quite difficult to follow. Given that it is currently quite long, authors need to edit the content carefully after considering the above comments. This could be achieved by shortening the introduction, avoiding repetitions between the tables and text in the results and trimming the discussion. 

Response: We edited the introduction for conciseness and shortened it. We also checked the text for repetition between the tables and text and removed any such cases. Finally, we modified the discussion and edited it for conciseness.

Parts of the manuscript also need better flow and focus. For example, why is the statement ‘The study adheres to the STROBE checklist’ included in the job satisfaction questionnaire section? The intent of mentioning ‘Basic Plan to Promote Cancer Control Programs’ is not clear. Analysis needs to be reviewed thoroughly and reanalyzed as necessary to be eligible for publication.

Response: We moved the statement regarding the STROBE checklist (page 10, lines 170-171). Furthermore, we clarified the intent of mentioning the Basic Plan to Promote Cancer Control Programs in the introduction (page 5, lines 75-77). We also reviewed the statistical analysis and conducted additional statistical analysis.

Additional comment:

The PLOS Data policy requires authors to make all data underlying the findings described in their manuscript fully available without restriction, with rare exception (please refer to the Data Availability Statement in the manuscript PDF file). The data should be provided as part of the manuscript or its supporting information, or deposited to a public repository. For example, in addition to summary statistics, the data points behind means, medians and variance measures should be available. If there are restrictions on publicly sharing data—e.g. participant privacy or use of data from a third party—those must be specified

Reviewer #2: Yes

Reviewer #3: No

Response: If the manuscript is accepted for publication, we will make the data fully available in a public repository.

---

## [Decision Letter · Decision Letter 2]

14 Apr 2020

Factors associated with the job satisfaction of certified nurses and nurse specialists in cancer care in Japan: Analysis based on the Basic Plan to Promote Cancer Control Programs

PONE-D-19-28473R2

Dear Dr. Arai,

We are pleased to inform you that your manuscript has been judged scientifically suitable for publication and will be formally accepted for publication once it complies with all outstanding technical requirements.

With kind regards,

Janhavi Ajit Vaingankar

Academic Editor

PLOS ONE

Additional Editor Comments (optional):

Reviewers' comments:

Reviewer's Responses to Questions

**Comments to the Author**

1. If the authors have adequately addressed your comments raised in a previous round of review and you feel that this manuscript is now acceptable for publication, you may indicate that here to bypass the “Comments to the Author” section, enter your conflict of interest statement in the “Confidential to Editor” section, and submit your "Accept" recommendation.

Reviewer #2: All comments have been addressed

Reviewer #3: All comments have been addressed

2. Is the manuscript technically sound, and do the data support the conclusions?

Reviewer #2: (No Response)

Reviewer #3: Yes

3. Has the statistical analysis been performed appropriately and rigorously? 

Reviewer #2: (No Response)

Reviewer #3: Yes

4. Have the authors made all data underlying the findings in their manuscript fully available?

Reviewer #2: (No Response)

Reviewer #3: Yes

5. Is the manuscript presented in an intelligible fashion and written in standard English?

Reviewer #2: (No Response)

Reviewer #3: Yes

6. Review Comments to the Author

Reviewer #2: (No Response)

Reviewer #3: My comments to the revisions have been adequately addressed. I would recommend proof reading the article to ensure there are no grammatical errors.

7. PLOS authors have the option to publish the peer review history of their article (what does this mean?). If published, this will include your full peer review and any attached files.

Reviewer #2: No

Reviewer #3: No

---

## [Editor Report · Acceptance letter]

1 May 2020

PONE-D-19-28473R2 

Factors associated with the job satisfaction of certified nurses and nurse specialists in cancer care in Japan: Analysis based on the Basic Plan to Promote Cancer Control Programs 

Dear Dr. Arai:

I am pleased to inform you that your manuscript has been deemed suitable for publication in PLOS ONE. Congratulations! Your manuscript is now with our production department. 

With kind regards,

on behalf of

Ms Janhavi Ajit Vaingankar 

Academic Editor

PLOS ONE